# A Validation of ERA5 Reanalysis Data in the Southern Antarctic Peninsula—Ellsworth Land Region, and Its Implications for Ice Core Studies

**Dieter Tetzner** [1,2,*], **Elizabeth Thomas** [1]  **and Claire Allen** [1]

1   British Antarctic Survey, High Cross, Madingley Road, Cambridge CB3 0ET, UK
2   Department of Earth Sciences, University of Cambridge, Downing Street, Cambridge CB2 3EQ, UK
*   Correspondence: dietet95@bas.ac.uk

**Abstract:** Climate reanalyses provide key information to calibrate proxy records in regions with scarce direct observations. The climate reanalysis used to perform a proxy calibration should accurately reproduce the local climate variability. Here we present a regional scale evaluation of meteorological parameters using ERA-Interim and ERA5 reanalyses compared to in-situ observations from 13 automatic weather stations (AWS), located in the southern Antarctic Peninsula and Ellsworth Land, Antarctica. Both reanalyses seem to perform better in the escarpment area (>1000 m a.s.l) than on the coast. A significant improvement is observed in the performance of ERA5 over ERA-Interim. ERA5 is highly accurate, representing the magnitude and variability of near-surface air temperature and wind regimes. The higher spatial and temporal resolution provided by ERA5 reduces significantly the cold coastal biases identified in ERA-Interim and increases the accuracy representing the wind direction and wind speed in the escarpment. The slight underestimation in the wind speed obtained from the reanalyses could be attributed to an interplay of topographic factors and the effect of local wind regimes. Three sites in this region are highlighted for their potential for ice core studies. These sites are likely to provide accurate proxy calibrations for future palaeoclimatic reconstructions.

**Keywords:** climate reanalysis; ERA5; ERA-Interim; Antarctic Peninsula; ice cores; Antarctica; winds

## 1. Introduction

In the last decades, the Antarctic Peninsula (AP) and West Antarctica (WA) have been experiencing dramatic climate changes [1]. Instrumental records suggest that since the 1950s, AP surface air temperatures are among the most rapidly warming on Earth [1,2]. This has been accompanied by ice shelf collapse and accelerated glacier flow, enhanced mass loss from melting glaciers [3] and increased snowfall [4–6], all having a direct and measurable impact on global sea levels [7,8]. Even though the AP and WA are undergoing particularly substantial environmental changes in our current era, they remain some of the more understudied regions on Earth.

Direct meteorological observations in the AP and WA are scarce, relatively short and mostly constrained to coastal regions. Only after the International Geophysical Year (1957–1958 A.D.), did stations begin to continuously and routinely record meteorological parameters, at a dozen sites along the coast of the northern AP. In addition to the station network, since the early 1980s, several automatic weather stations (AWS) and weather transmitters/GPS stations (hereinafter referred to as AWS) have been deployed in the region [9]. These stations have helped to improve the spatial coverage of meteorological observations, as well as provide data from remote locations. However, the region stretching between the southern AP and continental WA, the Ellsworth Land (EL) region, is still data sparse, hindering our ability to determine how far the climate extremes observed in the northern AP extend into the continent and to place these recent changes in a longer-term context [10].

Due to the lack of long-term direct observational data in the southern AP and EL region, the main sources of data for studying climate are the reanalysis datasets. Reanalyses compensate for the lack of in-situ data by using a wide range of observational data to constrain a full-physics atmospheric model, which provides the best estimate of the changing structure of the atmosphere through time [11]. Although climate reanalysis datasets can provide continuous climatic information extending back to 1979, they overlap with the period when the recent major climatic changes have been recorded, undermining their use to study long-term regional climate variability.

An alternative source of information about climate comes from ice cores, proven to yield valuable climatic information in this region through the use of proxy records [12–16]. To validate a proxy record, it must be first calibrated against accurate environmental measurements. In a region with few continuous long-term records, the reanalysis data can provide a key calibration tool. However, the extent to which reanalysis data captures local variability must first be assessed. This is because the reanalysis products are not exempt from uncertainties and biases from various sources that are inherent in the assimilation processes [17–19]. Previous studies on the reliability of global reanalysis in other regions of Antarctica have shown that ERA-Interim reanalysis present small, but measurable, biases [10,17,20–25]. Therefore, it is crucial to evaluate the regional reanalysis products against in-situ measurements, so they are made more reliable for when they are applied to climate studies.

In this study, we assess the accuracy of two climate reanalyses, ERA5 and ERA-Interim, to reproduce in-situ meteorological measurements obtained using a network of thirteen AWS deployed throughout the southern AP and EL region since 1982. The data obtained from these AWS offer the possibility to validate reanalyses against independent in-situ data which have not been assimilated into the reanalyses. The aim of this study is to determine which reanalysis captures more accurately the regional climate variability and to determine where are the best sites to retrieve ice cores to be compared to reanalysis data for proxy calibration.

## 2. Materials and Methods

### 2.1. Reanalysis Data

The ERA reanalyses, from the European Center for Medium-Range Weather Forecasts (ECMWF), are data assimilation systems that advance forward in time on cycles where available observations are combined with forecasts to deduce the evolving state of the atmosphere and the surface beneath it. The data assimilation produces a physically coherent database of the global state of the atmosphere constrained by available observations.

Data from the ECMWF ERA-Interim reanalysis [26], and from the recently released fifth generation of ECMWF reanalysis, ERA5 [27], were used to estimate surface air temperature (2 m temperature), wind speed (10 m wind speed), wind vector (10 m zonal wind (U) and 10 m meridional wind (V)) and precipitation (surface P-E parameter in ERA-Interim and surface total precipitation parameter in ERA5). ERA-Interim datasets are available at 0.7-degree resolution (~70 km), every six hourly intervals since 1979. Similarly, ERA5 reanalysis also extends back to 1979, but provides hourly data and at a much higher resolution (0.25 degree ~ 31 km) than ERA-Interim.

### 2.2. Observations from AWS

Figure 1 shows the location of the thirteen AWS used in this study and their nearest ERA-Interim and ERA5 grid points.

The thirteen AWS incorporated in this study recorded surface air temperature, atmospheric pressure, wind speed, and wind direction. Additionally, three of those AWS also recorded snow depth. All the datasets were subjected to quality control filters due to the presence of instrument failures, errors, gaps, and odd values. Single odd values, error labels, and data with zero values for both wind speed and direction were manually removed from the datasets. Once filtered, only the months with more than 50% of valid data (valid months) were taken into account for further analyses

and to calculate monthly mean values. Additionally, austral summer (December-January-February (DJF)) surface air temperatures recorded during wind events where speed remained below $2 \text{ m s}^{-1}$ ($\approx$16% of the summertime data) were removed to avoid temperature overestimations due to the lack of ventilation during low-wind speed conditions [24,28].

In order to avoid biases, this study only incorporates datasets from sites where all calendar months are represented. For this, a minimum threshold of 33% of data coverage was applied for each calendar month. Based on their geographical location, stations were classified either as coastal (<1000 m.a.s.l) or as escarpment (>1000 m.a.s.l) (Figure 1). The main parameters of each station and the distance to their respective nearest reanalysis grid points are summarized in Table 1.

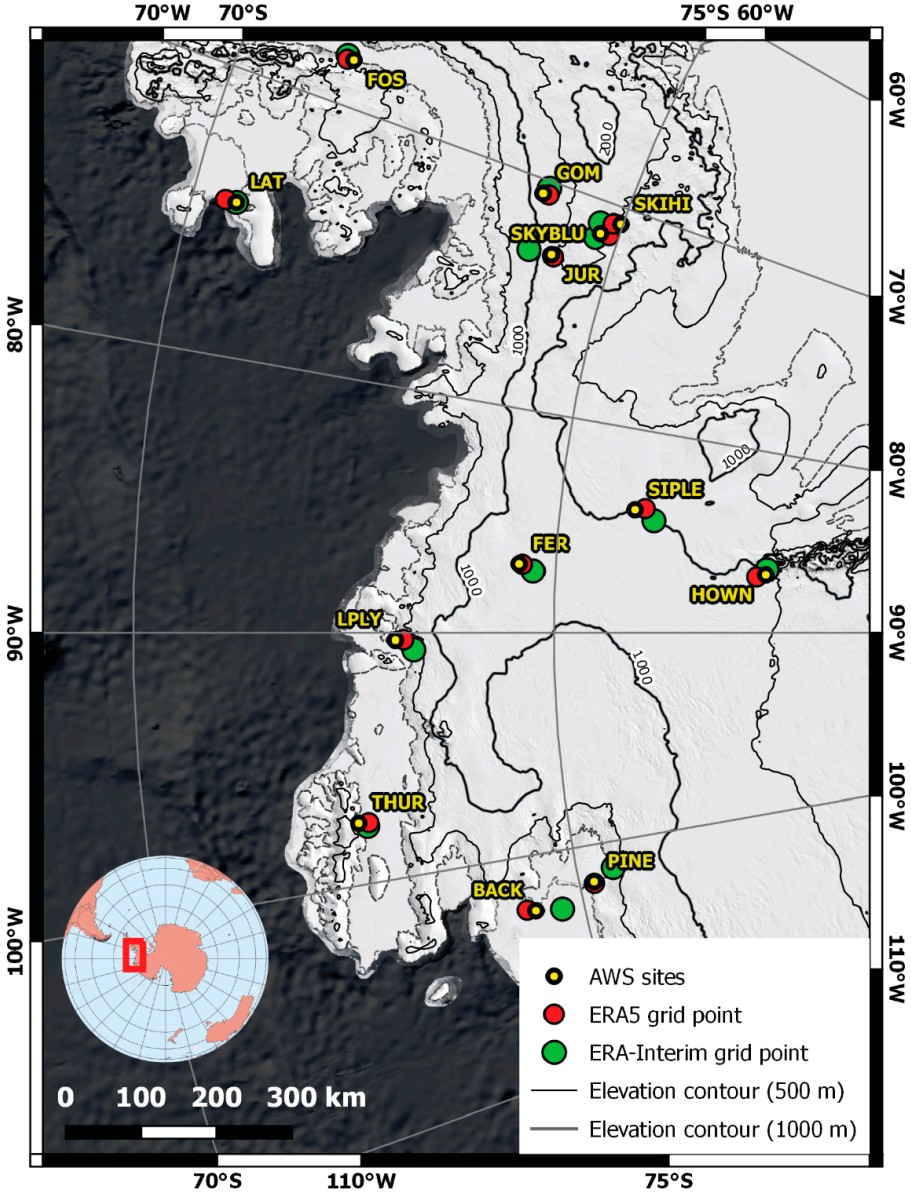

**Figure 1.** Map showing the automatic weather stations (AWS) sites considered in this study and their closest ERA-Interim and ERA5 grid points. The yellow circles show the locations of the thirteen AWS sites, Fossil Bluff (FOS), Latady Island (LAT), Gomez (GOM), Jurassic (JUR), Sky-blu (SKYBLU), Ski-Hi (SKIHI), Siple Station (SIPLE), Ferrigno (FER), Backer Island (BACK), Pine Island glacier (PINE), Thurston Island (THUR), Howard nunatak (HOWN), Lepley nunatak (LPLY).

**Table 1.** Summary of the AWS geographical location and main features of the AWS datasets. POLENET: Polar Earth Observing Network. BAS: British Antarctic Survey.

| Site | Acronym | Data Source (Online Database) | Lat (°S) | Long (°W) | Elevation (m a.s.l) | Data Interval (from–to) (mm/yy)(AD) | Number of Valid Months | Temporal Resolution (Minutes) | Distance to the Nearest Grid Point (km) | |
|---|---|---|---|---|---|---|---|---|---|---|
| | | | | | | | | | ERA-Int | ERA5 |
| **Costal** | | | | | | | | | | |
| Backer Island | BACK | POLENET | 74.43 | 102.48 | 38 | 01/12–12/16 | 48 | 30 | 34.76 | 11.4 |
| Fossil Bluff | FOS | BAS | 71.32 | 68.28 | 66 | 01/07–12/17 | 129 | 10 | 10.91 | 11.65 |
| Latady Island | LAT | BAS | 72.69 | 78.03 | 200 | 01/05–01/06 | 13 | 60 | 0.6 | 14.6 |
| Lepley Nunatak | LPLY | POLENET | 73.11 | 90.3 | 156 | 01/12–05/17 | 38 | 30 | 27.57 | 10.9 |
| Pine Island | PINE | U. of Wisconsin | 75.18 | 101.73 | 70 | 01/08–12/11 | 33 | 10 | 31.67 | 2.23 |
| Thurston Island | THUR | POLENET | 72.53 | 97.56 | 180 | 02/11–05/17 | 71 | 30 | 12.49 | 13.3 |
| **Escarpment** | | | | | | | | | | |
| Ferrigno | FER | BAS | 74.57 | 86.9 | 1376 | 01/10–12/10 | 12 | 10 | 20.7 | 4.56 |
| Gomez | GOM | BAS | 73.98 | 70.61 | 1400 | 02/05–06/06 | 17 | 60 | 10.64 | 7.69 |
| Howard Nunatak | HOWN | POLENET | 77.53 | 86.77 | 1478 | 01/10–05/17 | 89 | 30 | 7.17 | 12.3 |
| Jurassic | JUR | BAS | 74.3 | 73.05 | 1139 | 03/05–03/06 02/12–08/12 | 20 | 60 | 30.03 | 3.96 |
| Siple Station | SIPLE | U. of Wisconsin | 75.9 | 83.92 | 1054 | 01/82–04/92 | 89 | 180 | 28.89 | 13.4 |
| Sky-BLU | SKYBLU | BAS | 74.78 | 71.48 | 1556 | 01/00–12/17 | 142 | 10 | 8.43 | 11.8 |
| Ski-Hi | SKIHI | U. of Wisconsin | 74.98 | 70.77 | 1395 | 03/94–11/98 | 52 | 180 | 27.58 | 10.4 |

*2.3. Method to Evaluate Reanalysis Data*

The two reanalyses (ERA-Interim and ERA5) tested in this study were compared to the AWS data to evaluate their performance. The inputs from the reanalyses were the datasets provided by the nearest grid points to each AWS site. All datasets were converted to monthly means to directly compare in-situ observations to reanalyses. Consistently, months that were removed from AWS datasets during the filtering process, were also removed from the reanalyses.

This study incorporates three performance indicators to evaluate monthly mean values of surface air temperatures, wind speed and snow accumulation [21,29]: normalized bias (NBIAS), normalized mean absolute error (NMAE) and the normalized root-mean-square error (NRMSE), an indicator which effectively combines the errors of low correlation and high bias into one statistic [30]: $\text{NBIAS} = \frac{1}{N}\sum_{t=1}^{N}\frac{\bar{x}_{reanalysis}(t)-\bar{x}_{AWS}(t)}{\bar{y}_{AWS}}$, $\text{NMAE} = \frac{1}{N}\sum_{t=1}^{N}\left|\frac{\bar{x}_{reanalysis}(t)-\bar{x}_{AWS}(t)}{\bar{y}_{AWS}}\right|$, $\text{NRMSE} = \sqrt{\left(\frac{1}{N}\sum_{t=1}^{N}\left(\frac{\bar{x}_{reanalysis}(t)-\bar{x}_{AWS}(t)}{\bar{y}_{AWS}}\right)^2\right)}$, where $\bar{x}_{reanalysis}$ is the mean value for each month of the reanalysis, $\bar{x}_{AWS}$ is the mean value for each month of AWS measurements and $\bar{y}_{AWS}$ is the annual mean of the AWS measurements. Additionally, the Pearson's linear correlation (R) was included to test the statistical relationship between the reanalyses and the AWS measurements. All linear correlations presented in this study are statistically significant ($p < 0.05$) and were calculated using detrended data.

To compare surface air temperatures obtained from the reanalyses to the measurements from the AWS, a dry adiabatic lapse rate ($-9.8$ °C·km$^{-1}$) was considered in order to correct for the effects of elevation change between sites [22,30]. To compare the monthly precipitation estimates from the reanalyses to the snow depth measurements from the AWS, positive daily variations in the snow depth were summed for each month to determine the monthly precipitation on each site. Precipitation events were defined in the AWS snow depth record as days with snow depth variations higher than 1 cm. To compare total precipitation events recorded on each site by the AWS to reanalyses precipitation estimates, the minimum threshold defined for the AWS snow depth record was converted to its water equivalent assuming an Antarctic fresh snow density of 300 kg m$^{-3}$ [31]. Thus, a minimum threshold of 3 mm of daily precipitation was applied to define precipitation events on the reanalyses.

To convert the wind fields obtained from the reanalyses to the measurements from the AWS, we used the following equations: $W_{spd} = \sqrt{U^2 + V^2}$, $W_{dir} = tan^{-1}(-U, -V)x\frac{180}{\pi}$, where U is the zonal wind, V is the meridional wind, $W_{spd}$ is the wind speed and $W_{dir}$ is the wind direction. Additionally, to compare directly both datasets a height adjustment was applied to convert the 10 m AWS wind fields to 3 m. This adjustment was applied assuming a logarithmic wind profile with a roughness length of 0.001 m [32] and neutral atmospheric stability [33]. This adjustment is applied since the atmospheric stability is unknown from the AWS observations [24,33].

To fully test the wind reanalyses performances, the extended versions of the wind datasets (sub-daily resolution) were combined to produce seasonal rose plots to qualitatively compare the performance of winds on each reanalysis. As a convention, all wind directions are reported as the direction where the wind is coming from.

## 3. Results

The basic statistics of meteorological parameters for each AWS are classified in Table 2. Table 3 presents the seasonal temperature bias for each station. Table 4 displays the Pearson's linear correlation coefficients. Table 5 presents the statistics of the three performance indicators (NBIAS, NMAE, NRMSE).

**Table 2.** Summary of the statistics of meteorological parameters from AWS and climate reanalyses. Temp: mean surface air temperature (°C), Wspd: mean surface wind speed (m s$^{-1}$). Acc: mean snow accumulation (mm). Sdev: Standard deviation.

| Site | AWS | | | | | | ERA-Interim | | | | | | ERA5 | | | | | |
|---|---|---|---|---|---|---|---|---|---|---|---|---|---|---|---|---|---|---|
| | Temp | Sdev | Wspd | Sdev | Acc | Sdev | Temp | Sdev | Wspd | Sdev | Acc | Sdev | Temp | Sdev | Wspd | Sdev | Acc | Sdev |
| **Coast** | −11.82 | 3.27 | 6.8 | 2.1 | | | −13.02 | 1.77 | 5.92 | 1.36 | | | −12.33 | 2.13 | 5.33 | 1.72 | | |
| BACK | −10.91 | 6.78 | 6.95 | 1.61 | | | −11.99 | 6.62 | 6.04 | 1.2 | | | −10.81 | 6.78 | 5.12 | 1.02 | | |
| FOS | −8.91 | 6.1 | 3.6 | 0.85 | | | −10.62 | 5.23 | 4.3 | 0.89 | | | −10.38 | 5.63 | 3.03 | 0.61 | | |
| LAT | | | 6.16 | 1.61 | 98.91 | 60.12 | | | 4.96 | 1.15 | 62.03 | 23.17 | | | 4.12 | 0.55 | 91.5 | 48.66 |
| LPLY | −11.23 | 6.18 | 7.7 | 1.4 | | | −13.26 | 6.74 | 6.7 | 1.18 | | | −11.45 | 5.88 | 6.45 | 1.3 | | |
| PINE | −17.44 | 6.39 | 10 | 2.14 | | | −15.09 | 5.62 | 8.1 | 1.55 | | | −15.43 | 5.28 | 7.93 | 1.47 | | |
| THUR | −10.63 | 5.31 | 6.39 | 1.27 | | | −14.16 | 8.02 | 5.44 | 1.08 | | | −13.60 | 6 | 5.3 | 0.99 | | |
| **Escarpment** | −21.16 | 2.79 | 6.81 | 0.93 | | | −20.88 | 2.64 | 5.72 | 1.37 | | | −21.02 | 3.43 | 5.89 | 1.37 | | |
| FER | −24.83 | 6.51 | 8.53 | 1.9 | | | −23.34 | 6.02 | 8.53 | 1.57 | | | −23.98 | 6.22 | 8.57 | 1.55 | | |
| GOM | −17.18 | 5.04 | 6.13 | 0.81 | 71.08 | 45.46 | −19.6 | 5.43 | 5.45 | 0.69 | 72.61 | 29.90 | −18.54 | 4.91 | 5.94 | 0.76 | 91.17 | 46.18 |
| HOWN | −22.59 | 5.25 | 6.23 | 1.37 | | | −23.54 | 7.22 | 4.23 | 0.91 | | | −24.74 | 5.72 | 3.97 | 0.98 | | |
| JUR | −20.10 | 5.53 | 6.95 | 1.31 | 54.86 | 61.71 | −18.75 | 5.49 | 5.4 | 0.78 | 77.04 | 34.95 | −15.96 | 4.63 | 6.06 | 0.93 | 92.07 | 41.16 |
| SIPLE | −24.21 | 7.63 | 5.72 | 1.12 | | | −24.04 | 7.61 | 6.2 | 1.03 | | | −24.47 | 7.23 | 5.36 | 0.89 | | |
| SKYBLU | −19.71 | 5.73 | 7.3 | 2.03 | | | −19.09 | 5.77 | 5.15 | 0.94 | | | −20.46 | 5.39 | 5.66 | 1.62 | | |
| SKIHI | −19.50 | 6.48 | 6.79 | 1.97 | | | −17.81 | 6.05 | 5.1 | 0.9 | | | −18.98 | 5.71 | 5.65 | 1.39 | | |

**Table 3.** Seasonal temperature bias. DJF: December-January-February. MAM: March-April-May. JJA: June-July-August. SON: September-October-November.

| Site | ERA-Interim-AWS | | | | | ERA5-AWS | | | | |
|------|------|-----|-----|-----|-----|------|-----|-----|-----|-----|
| | Mean | DJF | MAM | JJA | SON | Mean | DJF | MAM | JJA | SON |
| **Coast** | −1.20 | −1.16 | −1.69 | −1.64 | −1.57 | −0.51 | −1.4 | −0.98 | −0.62 | −0.78 |
| BACK | −1.08 | −1.47 | −1.95 | −0.76 | −1.47 | 0.1 | −0.23 | −0.61 | −0.1 | −0.11 |
| FOS | −1.71 | −3.84 | −1.73 | −1.59 | −1.89 | −1.47 | −3.00 | −1.78 | −1.65 | −1.64 |
| LPLY | −2.03 | −1.43 | −2.61 | −1.87 | −2.64 | −0.22 | −0.84 | −0.13 | 0.87 | −0.07 |
| PINE | 2.35 | 2.72 | 3.33 | 4.08 | 2.74 | 2.01 | 0.95 | 2.25 | 2.89 | 1.99 |
| THUR | −3.53 | −1.79 | −5.51 | −8.05 | 4.59 | −2.97 | 3.90 | −4.63 | −5.11 | −4.05 |
| **Escarpment** | 0.28 | 0.67 | −0.1 | −0.09 | 0.38 | 0.14 | −0.4 | −0.03 | 0.11 | −0.14 |
| FER | 1.49 | 0.53 | −0.09 | 1.93 | 1.39 | 0.85 | 0.52 | 0.16 | 1.17 | 0.74 |
| GOM | −2.42 | −3.73 | −4.64 | −5.04 | −4.47 | −1.36 | −3.32 | −3.28 | −3.28 | −3.44 |
| HOWN | −0.95 | 5.31 | −0.31 | −0.19 | 1.75 | −2.15 | −1.2 | −3.26 | −3.23 | −2.55 |
| JUR | 1.35 | −1.81 | −0.76 | −2.58 | −0.9 | 4.14 | 4.63 | 6.59 | 5.9 | 5.99 |
| SIPLE | 0.17 | −0.13 | −0.29 | 0.01 | −0.06 | −0.26 | −1.33 | −0.73 | −0.35 | −0.85 |
| SKYBLU | 0.62 | 0.97 | 0.91 | 0.8 | 0.95 | −0.75 | −2.34 | −1.78 | −1.36 | −1.89 |
| SKIHI | 1.69 | 3.52 | 4.46 | 4.45 | 4.00 | 0.52 | 0.24 | 2.07 | 1.91 | 1.05 |
| **Total** | −0.34 | −0.1 | −0.77 | −0.73 | −0.43 | −0.13 | −0.82 | −0.43 | −0.2 | −0.4 |

**Table 4.** Pearson's linear correlation coefficient between monthly meteorological parameters and ERA-Interim reanalysis (A), and between monthly meteorological parameters and ERA5 reanalysis (B). All data presented are statistically significant ($p < 0.05$).

| Site | Temperature | | Wind Speed | | Zonal Wind (U) | | Meridional Wind (V) | | Snow Accumulation ** | |
|------|------|------|------|------|------|------|------|------|------|------|
| | A | B | A | B | A | B | A | B | A | B |
| **Coast** | | | | | | | | | | |
| BACK | 0.99 | 0.99 | 0.87 | 0.94 | 0.81 | 0.71 | 0.78 | 0.81 | | |
| FOS | 0.99 | 0.99 | 0.75 | 0.70 | * | 0.48 | 0.74 | 0.85 | 0.75 | 0.76 |
| LAT | | | 0.94 | 0.94 | * | * | 0.85 | 0.85 | | |
| LPLY | 0.98 | 0.97 | 0.74 | 0.77 | 0.70 | 0.7 | 0.66 | 0.57 | | |
| PINE | 0.97 | 0.98 | 0.9 | 0.95 | 0.79 | 0.77 | 0.52 | 0.72 | | |
| THUR | 0.95 | 0.98 | 0.76 | 0.89 | 0.74 | 0.75 | 0.65 | * | | |
| **Escarpment** | | | | | | | | | | |
| FER | 0.98 | 0.99 | 0.74 | 0.70 | 0.86 | 0.86 | 0.99 | 0.99 | | |
| GOM | 0.98 | 0.98 | 0.80 | 0.81 | 0.90 | 0.90 | 0.91 | 0.91 | 0.75 | 0.84 |
| HOWN | 0.95 | 0.98 | 0.79 | 0.84 | 0.72 | 0.73 | 0.69 | 0.62 | | |
| JUR | 0.97 | 0.96 | 0.55 | 0.61 | 0.81 | 0.82 | 0.78 | 0.84 | 0.8 | 0.81 |
| SIPLE | 0.99 | 0.99 | 0.80 | 0.84 | 0.94 | 0.94 | 0.93 | 0.92 | | |
| SKIHI | 0.99 | 0.99 | 0.78 | 0.95 | 0.66 | 0.66 | 0.89 | 0.96 | | |
| SKYBLU | 0.98 | 0.99 | 0.74 | 0.92 | 0.9 | 0.94 | 0.82 | 0.93 | | |

* values not statistically significant at 95% confidence level. ** Snow accumulation in ERA-Interim reanalysis is represented by the P-E parameter. Snow accumulation in ERA5 reanalysis is represented by the Surface precipitation parameter ("total precipitation").

## 3.1. Surface Air Temperature

Twelve AWS sites were considered for the surface air temperature evaluation. Table 2 shows that both reanalyses present a close agreement with the in-situ measurements. However, some significant differences were identified.

Mean surface air temperatures show a geographical segmentation in their distribution through the study region. Coastal sites present higher mean temperatures while the escarpment sites are characterized by lower mean temperatures. Both reanalyses reproduce this segregation. However, mean surface air temperature values reported by the reanalyses are slightly different from the mean in-situ measurements. Both reanalyses tend to overestimate the mean temperatures in the coast (ERA-Interim: −1.2 °C, ERA5: −0.51 °C), while they tend to underestimate them in the escarpment area

(ERA-Interim: +0.28 °C, ERA5: +0.14 °C). In particular, the overestimation in the coast is greater than the underestimation in the escarpment. From a seasonal perspective, in both reanalyses, all seasons present cold biases. The austral summer (DJF) is highlighted for presenting the highest seasonal bias values in ERA5 (in the escarpment and coastal areas). Similarly, during DJF, ERA-Interim presents the highest bias value in the escarpment, but the lowest bias on the coast. Temperature bias values exhibit small variations during the rest of the year. Overall, mean surface air temperatures estimated from ERA5 present a smaller mean bias value than the one obtained from ERA-Interim. Additionally, ERA5 does not present a clear geographical bias pattern, as observed in ERA-Interim. The mean absolute error in the escarpment area presents similar values in both reanalyses (ERA-Interim: 1.24, ERA5: 1.43). However, on the coast, the mean absolute error is larger in ERA-Interim (ERA-Interim: 2.14, ERA5: 1.35).

The analyses carried out using mean values of the performance indicators and correlation coefficients report significant variations of ERA5 versus ERA-Interim. Comparatively, ERA5 shows higher linear correlation coefficients, accompanied by a decrease of 2% in the mean NBIAS and mean NMAE, and 4% in the mean NRMSE. Geographical segmentation is also present in these analyses. In both reanalyses, mean performance indices values in the escarpment area tend to be lower than the ones obtained in the coastal area. The lowest performance index (NBIAS, NMAE, and NRMSE) and highest correlation coefficient's (R) are in ERA5 datasets of escarpment stations: SIPLE and SKYBLU. Nevertheless, the greatest single station decreases, from ERA-Interim to ERA5, were recorded in the coastal area. The greatest decrease in NBIAS and NMAE was recorded in LPLY (16% and 9%, respectively), while for NRMSE, it was recorded in THUR (14%).

### 3.2. Snow Accumulation

Three AWS sites were used to evaluate the snow accumulation in the region. Mean snow accumulation values present significant differences from site to site. The highest value corresponds to the only coastal site (LAT). There, the magnitude estimated by ERA5 shows the closest agreement with the observations. Conversely, ERA-Interim exhibits a higher accuracy predicting the magnitude of the mean snow accumulation in the escarpment area (GOM and JUR).

Linear correlation coefficients show consistently higher coefficients in ERA5 datasets. The highest correlation coefficients are found in the escarpment area, where a significant increase is documented in the ERA5 dataset at GOM site (ERA-Interim R = 0.75, ERA5 R = 0.84). The performance index shows three opposed scenarios with no clear trend.

Both reanalyses underestimate the occurrence of precipitation events recorded in the AWS. A total of 181, 209 and 133 precipitation events were identified in GOM, JUR, and LAT, respectively. Total precipitation events identified in ERA5 represent 86%, 89%, and 85%, of the total events identified in the AWS record of GOM, JUR and LAT, respectively. Likewise, total precipitation events identified in ERA-Interim account for 78%, 71%, and 89% of the total events identified in the AWS record of GOM, JUR and LAT, respectively.

**Table 5.** Performance indicators obtained using AWS datasets and climate reanalyses. Temp: Temperature. Wspd: Wind speed. Acc: Accumulation.

| Site | NBIAS (%) | | | | | | NMAE (%) | | | | | | NRSME (%) | | | | | |
|---|---|---|---|---|---|---|---|---|---|---|---|---|---|---|---|---|---|---|
| | ERA-Interim vs. AWS | | | ERA5 vs. AWS | | | ERA-Interim vs. AWS | | | ERA5 vs. AWS | | | ERA-Interim vs. AWS | | | ERA5 vs. AWS | | |
| | Temp | Wspd | Acc | Temp | Wspd | Acc | Temp | Wspd | Acc | Temp | Wspd | Acc | Temp | Wspd | Acc | Temp | Wspd | Acc |
| **Coast** | 0.14 | −0.1 | | 0.07 | −0.21 | | 0.2 | 0.17 | | 0.15 | 0.22 | | 0.26 | 0.21 | | 0.18 | 0.25 | |
| BACK | 0.11 | −0.13 | | −0.01 | −0.26 | | 0.14 | 0.14 | | 0.08 | 0.26 | | 0.22 | 0.17 | | 0.12 | 0.28 | |
| FOS | 0.19 | 0.2 | | 0.17 | −0.16 | | 0.2 | 0.22 | | 0.17 | 0.18 | | 0.24 | 0.27 | | 0.19 | 0.22 | |
| LAT | | −0.19 | −0.37 | | −0.33 | −0.07 | | 0.19 | 0.46 | | 0.33 | 0.2 | | 0.22 | 0.58 | | 0.37 | 0.26 |
| LPLY | 0.18 | −0.13 | | 0.02 | −0.16 | | 0.2 | 0.14 | | 0.11 | 0.17 | | 0.23 | 0.18 | | 0.12 | 0.2 | |
| PINE | −0.12 | −0.19 | | −0.1 | −0.2 | | 0.13 | 0.19 | | 0.13 | 0.2 | | 0.18 | 0.21 | | 0.16 | 0.22 | |
| THUR | 0.33 | −0.14 | | 0.28 | −0.17 | | 0.33 | 0.16 | | 0.28 | 0.17 | | 0.44 | 0.2 | | 0.3 | 0.19 | |
| **Escarpment** | −0.01 | −0.16 | | 0 | −0.14 | | 0.09 | 0.2 | | 0.08 | 0.27 | | 0.1 | 0.24 | | 0.09 | 0.19 | |
| FER | −0.06 | 0 | | −0.03 | 0 | | 0.07 | 0.09 | | 0.04 | 0.1 | | 0.08 | 0.1 | | 0.05 | 0.12 | |
| GOM | 0.14 | −0.11 | 0.04 | 0.08 | −0.03 | 0.19 | 0.14 | 0.12 | 0.32 | 0.08 | 0.8 | 0.32 | 0.15 | 0.14 | 0.41 | 0.09 | 0.1 | 0.41 |
| HOWN | 0.04 | −0.32 | | 0.1 | −0.36 | | 0.12 | 0.32 | | 0.12 | 0.36 | | 0.15 | 0.34 | | 0.13 | 0.38 | |
| JUR | −0.07 | −0.22 | 0.41 | −0.2 | −0.13 | 0.69 | 0.12 | 0.23 | 0.63 | 0.2 | 0.16 | 0.85 | 0.14 | 0.27 | 0.81 | 0.24 | 0.19 | 0.95 |
| SIPLE | −0.01 | 0.09 | | 0.01 | −0.06 | | 0.02 | 0.12 | | 0.02 | 0.09 | | 0.02 | 0.15 | | 0.03 | 0.12 | |
| SKYBLU | −0.03 | −0.29 | | 0.04 | −0.22 | | 0.04 | 0.3 | | 0.04 | 0.22 | | 0.04 | 0.35 | | 0.05 | 0.24 | |
| SKIHI | −0.09 | −0.25 | | −0.03 | −0.17 | | 0.09 | 0.25 | | 0.06 | 0.17 | | 0.11 | 0.33 | | 0.07 | 0.21 | |

*3.3. Winds*

Thirteen AWS sites were considered to evaluate the winds in the study region. Mean wind speed values measured in-situ show that there is no clear geographical distribution. Sites with higher wind speed values are scattered throughout the region (coast: PINE and LPLY, escarpment: FER and SKYBLU).

Mean wind speeds estimated from both reanalyses are in close agreement with the observations from AWS sites. However, their accuracy varies across the region and both tend to underestimate the mean wind speed (ERA-Interim: $-1.0$ m s$^{-1}$, ERA5: $-1.18$ m s$^{-1}$). ERA-Interim tends to underestimate the mean wind speed in the escarpment, but it provides more accurate predictions in the coastal area. While ERA5 underestimates the mean wind speed in the coastal area, it gives accurate predictions at the escarpment. Overall, the mean ERA5 bias in the coastal zone ($-1.48$ m s$^{-1}$) is higher than mean ERA-Interim bias at the escarpment ($-1.08$ m s$^{-1}$). Conversely, in the escarpment, ERA5 presents the smallest mean absolute error (0.93 m s$^{-1}$).

Wind speed correlation coefficients show a consistent increase in the ERA5 datasets. ERA5 correlation coefficients are generally higher on the coast. However, the highest correlations (R = 0.95) are found in both areas (PINE and SKIHI). Correlation coefficients calculated using the wind vector components (U and V) exhibit different behavior. In the entire study region, there is no clear agreement on which reanalysis presents higher correlation values. Nevertheless, the V component shows a considerably higher variability in the coefficients obtained from each reanalysis, compared to the U component. Spatially, both components agree that the escarpment area presents higher correlation coefficients (R > 0.62) and that the higher coefficients are associated with ERA5 datasets. The highest correlation values for the U component are in SIPLE (R = 0.94) and SKYBLU (R = 0.94), while for the V component are in SKYBLU (R = 0.96) and FER (R = 0.99). On the other hand, mean correlation coefficients in the coastal area show comparatively small variations between reanalyses. Additionally, some sites in the coastal area show values which are not statistically significant ($p > 0.05$) (values labeled as "*" in Table 4).

Mean values of the performance indicators show that even though ERA5 datasets increase the NMAE and NBIAS by 4% and 6%, respectively, NRMSE decreases by 1%. In particular, the ERA5 mean NRMSE reports a 5% decrease in the escarpment and a 4% increase on the coast. Two stations are consistent in presenting the smallest performance values in the region (THUR and FER). Overall, NBIAS, NMAE, and NRMSE do not exhibit a clear geographical segmentation in the wind parameter.

A qualitative analysis of multiyear seasonal wind rose plots shows how reanalyses perform in reproducing the main wind direction and seasonality (Figure 2, not all sites are shown). In most of the sites (11 out of 13), the reanalyses datasets present a close agreement with the main measured wind direction ($\pm30°$), the main direction-speed frequency ($\pm5\%$) and the seasonal distribution. Only two sites located in the southern AP coast show no agreement between reanalyses data and in-situ measurements (LAT and FOS). The wind rose plots also indicate that both reanalyses present difficulties in accurately estimating high wind speed, which is generally underestimated by the reanalyses. Additionally, five sites present more than one preferential wind direction, which is not always completely represented by the reanalysis wind rose plots. A qualitative comparison between the reanalyses wind rose plots shows higher accuracy in ERA5 wind representations.

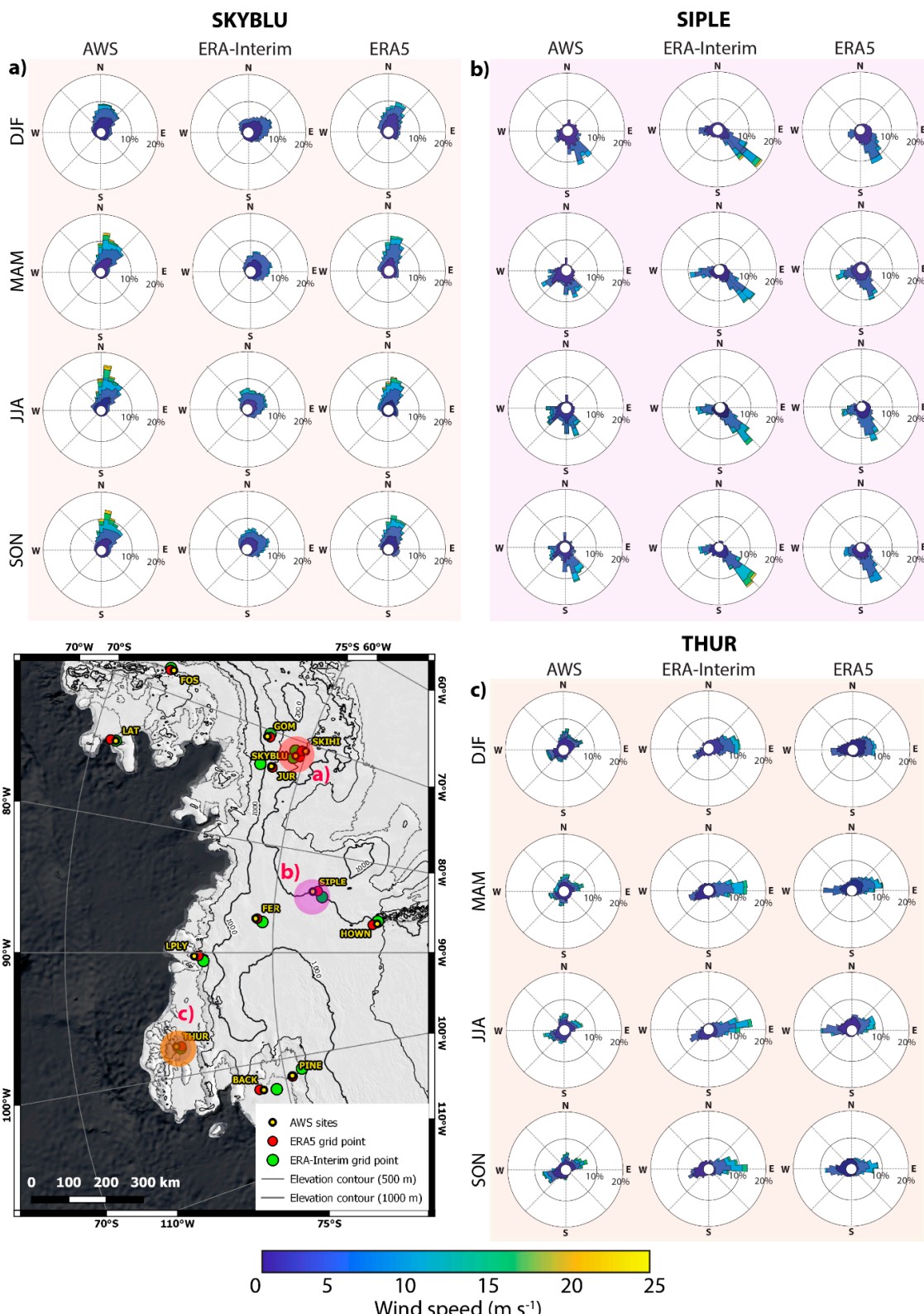

**Figure 2.** Seasonal wind rose plots on three AWS sites and their location within the study region. (**a**) Seasonal wind rose plots of SKYBLU AWS (2000–2017). (**b**) Seasonal wind rose plots of SIPLE AWS (1982–1992). (**c**) Seasonal wind rose plots of THUR AWS (2011–2017).

## 4. Discussion

### 4.1. Climate Reanalyses Performance

In the southern AP and EL region, climate reanalyses have shown different levels of performance depending on the meteorological parameters and the geographical context. Surface air temperature is the parameter that is best captured by both reanalysis across the whole region (R = 0.98, $p < 0.05$). The variations between the in-situ measurements and the reanalyses predictions could be partly attributed to differences in elevation and the inland distance between the AWS and the reanalyses grid points. Even though surface air temperatures were corrected considering a uniform dry adiabatic lapse rate, it has been shown that lapse rates are likely to be highly variable throughout the year, due to surface-based temperature inversions on the margins of Antarctica [34–36]. Thus, it should be acknowledged that a standard correction may introduce new errors because of the assumptions taken. Additionally, a minor, but a potential source of variability, could be associated with variations on the height of the temperature measuring instrument due to partial station burial throughout the year. Either way, our results show that in both reanalyses mean surface air temperatures are cold biased in the coastal (ERA-Interim:−1.2 °C, ERA5:−0.51°C) area, but slightly warm biased at the escarpment (ERA-Interim:+0.28°C, ERA5:+0.14 °C). These findings corroborate the cold bias (between −1.1 and −3 °C) obtained in ERA-Interim datasets from previous studies in the western-AP, Amundsen Sea Embayment and Ross Sea coastline [23,24]. At the same time, the findings agree with the warm bias documented in high elevation sites (>1000 m a.s.l.) in ERA-Interim datasets in East Antarctica [17,23]. Conversely, our results contradict the high-altitude cold bias obtained in previous attempts to validate the ERA-Interim dataset in the Ellsworth Land region based on two AWS stations (FER and JUR) [10]. These differences may arise from the fact that our study incorporates longer time series and data filtering assumptions that were not considered by Thomas and Bracegirdle (2015). Overall, ERA5 mean temperature estimations more accurately reproduce the observed temperature variability.

Snow accumulation measurements are highly underrepresented in the study region. The scarcity of data leads to contrasting scenarios where it is difficult to identify which reanalysis performs better. In particular, the mean magnitude of the snow accumulation parameter in the escarpment is better resolved by the ERA-Interim, while the ERA5 improves the accuracy of the meteorological variability. This could be due to the fact that ERA-Interim and ERA5 do not strictly estimate the same parameter. ERA-Interim estimates the combined effect of precipitation and evaporation in the surface layer (P-E), while ERA5 gives an estimation of the total precipitation on the surface. Either way, both reanalyses should present a bias because they ignore the effect of wind removal and re-deposition of snow, which is not negligible in Antarctica [37–40].

The new ERA5 reanalysis has proven to be highly accurate at identifying precipitation events in this region (recognizing over 85% of the events measured by the AWS). The absence of some precipitation events in the reanalyses estimations could be partly attributed to the inaccuracies in the magnitude of the precipitation estimates, possibly sometimes estimating precipitation events below the minimum threshold. Despite this slight difference, the remarkable ability of ERA5 to capture the occurrence of precipitation events in this region will significantly contribute to the future study of ice core proxies. Particularly, it will allow tracking of the pathways followed by the air masses producing precipitation and to characterize the source regions of marine and continental species deposited at the ice core sites [10]. However, our results highlight the need for a larger amount of long-term snow accumulation measurements to accurately evaluate the reanalyses performance.

Previous studies have assessed the performance of various reanalysis products with snow accumulation and surface mass balance over regional and continental scales [25]. However, only two studies have evaluated the wind parameter over regional and continental scales [21,24], presenting a large gap in the southern AP and EL region. Moreover, none have specifically evaluated the wind vector in the reanalysis data. Our study fills this gap and suggests that winds have been accurately estimated in most of this region. The variability of the accuracy on each site could be explained by

an interplay of topographic factors and the effect of local wind regimes. The sites which are better estimated are located in the escarpment areas, where topography promotes winds flowing downslope following a constant direction. These sites also favor the development of stable wind conditions which are better captured by the climate reanalyses. Conversely, sites that are not effectively captured by the reanalyses are located in coastal areas of the AP, areas which are characterized by complex topography and continuously subjected to synoptic variations.

Wind speed variability is accurately estimated by the reanalyses. However, both reanalyses appear to slightly underestimate wind speed magnitude. This is particularly relevant when determining the influence of blowing snow, a key unknown in surface mass balance measurements from ice cores. The high-speed biases obtained from the reanalyses can be explained by a combination of two factors. The first factor is the difference in the dataset recording frequency. The sub-hourly resolution of most of the AWS contrasts with the hour to multi-hour resolution of the reanalysis products. The high-resolution measurements from the AWS record more frequently high wind speed events (e.g., wind gusts), increasing the variability of the observational record and biasing the mean values towards higher speeds. The second factor is attributed to the local surface intensification of winds in the escarpment and coastal areas. This intensification can be observed in katabatic confluence zones, where the local topography channels the surface flow, generating very high surface wind speeds [9]. Previous studies have shown that this katabatic effect is particularly strong in the escarpment and coastal areas around Antarctica [41] and that it causes significant biases in the reanalyses estimations [21]. The mean wind speed bias obtained from ERA-Interim-ERA5 in this study ($-1.09$ m s$^{-1}$) agrees with the ERA-Interim annual mean bias previously estimated in the Amundsen Sea Embayment ($-1.32$ m s$^{-1}$) [24]. An additional, but likely small, source of bias could be attributed to the assumption of neutral atmospheric stability and a constant elevation of the AWS above the surface. The 10 m logarithmic wind profile adjustment performed in this study did not consider the possible effect of partial AWS burial. A decrease in the height of the instruments measuring the wind would slightly decrease the magnitude of the winds measured by the AWS and would imply reconsidering the height to adjust the vertical wind profile. Thus, considering the effects of a partially buried AWS would increase the bias estimation reported in this study.

The three meteorological parameters analyzed are consistent in presenting a significant improvement of ERA5 compared to ERA-Interim. These improvements could be associated with the higher spatial and temporal resolution provided by ERA5. A higher resolution helps the reanalysis to produce a more realistic topography, as well as to capture local and shorter meteorological variations that are effectively measured by the AWS. Our results match with previous studies in Antarctica showing that an increase in the spatiotemporal resolution leads to a better performance of the reanalysis [11,17,21].

Spatially, the meteorological parameters agree with a better performance of ERA5 in the escarpment area (>1000 m.a.s.l). The escarpment presents a relatively uniform surface compared to the coastal area where the combination of rough topography, variable ocean-ice-atmosphere thermal interactions, and a higher dependence on synoptic activity lead to higher climate variability.

Our results highlight two AWS sites at the coast (THUR and PINE) and two at the escarpment (SIPLE and the combined SKYBLU-SKIHI), where ERA5 datasets show the greatest potential to accurately reproduce the meteorological variability at each site.

## 4.2. Implications for Ice Core Proxy Calibration

The reanalysis evaluation has demonstrated that ERA5 efficiently captures the meteorological variability of the study region. Even though the ERA5 has been validated, subtle variations exist on its accuracy from site to site. To ensure the highest accuracy in the region, the vicinities of the previously highlighted AWS can be targeted. These sites are clear candidates to provide precise long-term meteorological records which enable the calibration of ice core proxies. However, not all of them present the optimal conditions to drill ice cores, from a climatological perspective. A suitable

place should meet specific conditions to ensure that the ice core contains a proxy and is preserved through time (e.g., temperatures below the freezing point, the source of air masses, etc.). Additional non-climatic parameters should be taken into account when selecting an ice core site, such as the regional geothermal heat flux, local elevation changes, the ice thinning process and ice flow dynamics. To determine the most appropriate location for an ice core survey, each site will be discussed separately.

The PINE station is located at a low elevation site (70 m a.s.l.) with the highest mean wind speed, among the AWS considered for this study. Even though this site presents a significant amount of annual snow accumulation (~750 mm w.eq.) [42], the low elevation exposes the location to surface melting and refreezing, affecting the preservation of the proxies. Additionally, the strong downslope winds are shown to persist from inland Antarctica, possibly limiting the use of this site for oceanic wind reconstructions. Even though the reanalyses have demonstrated to accurately estimate the meteorological variability at PINE, this site is not completely suitable for ice core studies.

The combined SKYBLU-SKIHI site presents the longest meteorological record in the region (194 months of valid observations). Our results largely validate the reanalyses from this site. Its high elevation (1556 m a.s.l.) prevents the occurrence of surface melting events, while its snow accumulation (~0.3 meters per year [43]) ensures enough snow to get a significant number of samples per year. Winds at this site have a consistent direction throughout the year (~ from North to South) (Figure 2), and a wide range of wind speeds. Additionally, backward trajectories analyses in the near JUR site show that air masses that reach this area come mostly from the Amundsen−Bellingshausen Sea [10]. A near oceanic source for the air masses provides several compounds and particles that after being transported, precipitated and buried on the snow, have the potential to become proxies. This, together with the constant unimodal wind direction at this site, provides an ideal background to develop a meridional wind reconstruction using ice cores. All these together make SKYBLU-SKIHI a promising location for ice core studies.

The THUR site is a low elevation site (180 m a.s.l.) located between Thurston Island (TI) and Sherman Island (SI). This site holds one of the longest meteorological records along the coast (71 months of valid observations), effectively validating the reanalyses in the surrounding area. The low elevation of this site exposes it to the formation of melt layers in the surface during the austral summer season, thus affecting the preservation of proxies on the ice. However, the higher elevation of the surrounding SI (~460 m a.s.l.) highlights it as a suitable place to retrieve a well-preserved ice core. SI is located in a strategic place as the higher elevation TI (~800 m a.s.l) shields it from meridional winds and from synoptic storms in the Amundsen−Bellingshausen Sea. Even though SI is under an effective rain shadow effect, a significant amount of snow is deposited every year (~600–700 mm w.eq.) [42]. Winds in this area are particularly affected by the local topography. The topography channelizes the air flowing parallel to the coast and produces a year-round bimodal wind pattern which is aligned with the zonal winds. In addition, the proximity of this area to the ocean ensures that a significant flux of chemical compounds and particles will be deposited on the snow that precipitates on top of these islands, providing potential environmental proxies. The synoptically driven zonal wind direction and the well constrained meteorological conditions on this site offers the possibility to track changes or biases in the input of the proxies due to changes in the air mass direction. Overall, the vicinities of THUR site present a high potential for ice core studies.

Among the targeted locations, SIPLE is the only one where an ice core has been retrieved [12]. The Siple Station ice core was drilled in 1986 and contains proxy records that extend back to 1430 A.D. This site presents suitable glaciological conditions for ice core studies and presents an average annual snow accumulation of ~0.56 meters per year [12]. Proxy records obtained from this ice core show similar, but shorter and less pronounced, trends to the ones observed in ice cores from the AP [44]. However, the onset of these trends is only a couple of years before the ice core was drilled, hindering the possibility to test their significance. Moreover, the ice core record only overlaps with the satellite era for seven years (1979–1985 A.D), limiting the proxy calibration period. A new shallow depth ice core from this site will enable an extension of the proxy calibration period by three decades. Additionally,

it will make it possible to test if the proxies on the ice have recorded the increase in temperatures and precipitation that the ERA5 reanalysis estimates for this site.

## 5. Conclusions

The ability of ERA-Interim and ERA5 reanalyses to capture the meteorological variability in the AP and EL regions was evaluated using thirteen AWS in situ datasets with measurements covering between 1982 and 2017. Both reanalyses present biases in the surface air temperatures. Significant cold bias was identified in ERA-Interim (−1.2 °C) on the coast, while a slight warm bias was identified in ERA5 (+0.14 °C) in sites at a higher elevation. Likewise, both reanalyses are not completely accurate in estimating the winds. Both reanalyses underestimate the wind speed with a higher bias for ERA5 in coastal regions (−1.48 m s$^{-1}$) and higher for ERA-Interim (−1.08 m s$^{-1}$) in sites at greater elevations (>1000 m a.s.l.). Additionally, the prevailing wind regimes have shown to be better represented by the ERA5 reanalysis. The surface wind evaluation conducted in this study is the first of its kind in the region and fills a gap in previous regional and continental-scale studies.

There is a significant improvement in the representation of both meteorological parameters when there is a higher spatial and temporal resolution. The improvement of ERA5 versus ERA-Interim in surface air temperatures and wind speed is significant with a decrease of 4% and 1% in the mean NRMSE and an increase of 0.01 and 0.06 in the mean correlation coefficient, respectively.

Overall, ERA5 presents a significant improvement compared to ERA-Interim. The best performance of ERA5 is at higher elevation areas (>1000 m a.s.l.), where the reanalysis reproduces a more realistic topography than in coastal areas.

Three sites are highlighted in the AP and EL region for their potential to provide valuable ice core proxy records which can be accurately calibrated with the ERA5 reanalysis data. These records will provide key information to determine how this region has been affected by the recent environmental changes observed in the northern AP.

**Author Contributions:** Conceptualization, D.T. and E.T.; methodology, D.T.; formal analysis, D.T., E.T. and C.A.; investigation, D.T.; Writing—Original Draft preparation, D.T.; Writing—Review and Editing, D.T., E.T. and C.A.

**Funding:** This research was funded by CONICYT–Becas Chile and Cambridge Trust funding program for PhD studies. Grant number 72180432.

**Conflicts of Interest:** The authors declare no conflict of interest. The funders had no role in the design of the study; in the collection, analyses, or interpretation of data; in the writing of the manuscript, or in the decision to publish the results.

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
