# Peer review of "A Validation of ERA5 Reanalysis Data in the Southern Antarctic Peninsula—Ellsworth Land Region, and Its Implications for Ice Core Studies"

_geosciences, doi:10.3390/geosciences9070289_

Reviewer 1 Report

Review of Tetzner et al., 2019, Geosciences

31 May 2019

Tetzner and co-authors provide a statistical assessment of ERA5 and ERA-Interim reanalysis products, in relation to an array of automated weather stations. They focus on the region near the base of the Antarctic Peninsula and Ellsworth Land. They present the analysis clearly and make a valuable contribution to future paleoclimate research in this part of Antarctica by showing which climate parameters are best represented by the reanalysis products in which portions of this region. I suggest some changes below which I hope will strengthen and clarify the paper.

My main suggestions relate to the methods. With the data coverage, representation of a month at least once seems like a low threshold. What does once represent? Maybe instead use a threshold percentage of potential readings for a given month. Or at least quantify the data coverage for the stations so it is clear. I also wonder about how different data recording frequencies (e.g., 10 minutes vs. 6 hours) influence the end result after converting the data to monthly means. It seems like higher-resolution measurements (e.g., AWS) would be able to record more high wind-speed events, whereas lower-resolution reanalysis products will always tend to underestimate higher wind speeds. The paragraph beginning on line 345 presents plausible reasons for the underestimate of high wind speeds by the reanalyses. But I’d also like to see a more thorough treatment of the raw data that go into the monthly means. For instance, it may be most appropriate to resample the AWS data to achieve a lower resolution prior to calculating the monthly mean value, so that the data more closely match the reanalysis products. I suspect this would result in wind speed values that more closely match the reanalysis values.

Specific comments:

“Escarpment” delineation is unclear even though defined. Needs to be shown on map and highlighted in Table 1 (as it is in subsequent tables). I’d also suggest defining it briefly in the abstract.

Fig. 1: add some numbers to elevation contours, make 1000 m contour darker to distinguish coastal from escarpment sites.

Line 155: Ref. 31 doesn’t seem like the most appropriate reference because it describes snow on sea ice. A density of 300 kg/m3 is quite low for West Antarctica. I’d suggest using a more geographically representative citation for these calculations, such as the recent paper by Fegyveresi et al. (2018). For instance, a density of 400 kg/m3 would be more reasonable.

Line 166: This is not correct. The convention is the other way around: wind rose frequency plots indicate the direction the wind is coming from. Please correct the plots and text accordingly.

Table 3: P value typically not reported as a percentage.

Lines 196-198: Use of “overestimation” and “underestimation” seems incorrect. Shouldn’t it be the opposite? Negative values should correspond with underestimation. This would be consistent with the idea of the cold coastal bias presented earlier and described in previous studies.

Are there any seasonal differences in the temperature biases? Please describe in this section.

Line 200: “ERA5 does” not “do.”

Lines 201-203: Where do the “mean absolute bias” values come from? Why are these different than the values listed earlier in the paragraph?

Section 3.2: I am having trouble following the logic in this section. First we learn that ERA-Interim is more accurate at the higher-elevation sites, then we learn that ERA5 correlation coefficients are consistently higher. Please clarify.

Throughout the manuscript, “reanalysis” is used both in the singular and in the plural when it should be “reanalyses.” Likewise, “reanalyses” is sometimes used when it should be the singular, “reanalysis.” Please read through carefully for correct usage.

Lines 308-309: Please revise grammar.

Lines 333-334: There is a published comparison of ERA-Interim and AWS data made at the Kominko-Slade station (WAIS Divide) by Koffman et al. (2014) which may be relevant here.

Line 353: Please clarify whether you mean the ERA-Interim or ERA5 wind speed bias here.

Line 359: Please revise fragment.

Section 4.2: Overall I think this is a useful assessment for future ice core studies. However, the implication that coastal sites have more proxies than inland sites is not supported by the vast literature describing paleoclimate interpretations from chemical proxies all across Antarctica. Certainly there are concentration gradients from the coast going inland. However, even the most remote, highest-elevation sites contain enough impurities in the snow to provide highly meaningful proxy interpretations. If there is a proxy specific to coastal sites that you can reference here (e.g., Lines 402 and 418), that would be helpful. Otherwise, this general statement about coastal sites having more chemical proxies is not really valid. I suggest revising the wording in this section.

References cited:

Fegyveresi, J.M. et al. (2018). Surface formation, preservation, and history of low-porosity crusts at the WAIS Divide site, West Antarctica. The Cryosphere 12, 325-341, doi: 10.5194/tc-12-325-2018.

Koffman, B. G. et al. (2014). Centennial-scale variability of the Southern Hemisphere westerly wind belt in the eastern Pacific over the past two millennia. Climate of the Past 10, 1125-1144, doi:10.5194/cp-10-1125-2014.        

Author Response

Our response to the reviewer's comments are in the attached file "author-coverletter-4344706.v1.docx"

Reviewer 2 Report

This paper show that a regional scale evaluation of meteorological parameters using ERA-Interim and ERA5 reanalyses compared to in-situ observations from 13 automatic weather stations (AWS), located in the southern Antarctic Peninsula and Ellsworth Land, Antarctica.

Both reanalyses show to perform better in the escarpment area, than on the coast. You found that three sites were likely to provide accurate proxy calibrations for future palaeoclimatic reconstructions.

If you can estimate the observational errors of temperature, wind speed and snow accumulation AWS, please describe them. How should these errors be evaluated when using monthly averages?

With regard to air temperature and wind speed, if the observed height of sensor changes, it will be more than the deviation between AWS and ERA being discussed.

You are trying to match AWS observational values to the height of ERA. With regard to air temperature, there is often a surface inversion layer on glacier. The wind speed also changes its slope depending on the degree of atmospheric instability. Since there is little evaluation about these height problems, ambiguity remains in discussion.

You point out that ERA is an effective method to search for sites that can reconstruct the paleoclimate from ice core. But logic seems to be weak.

L100:There is a quality control filter in the data of AWS. Did it manually?

L103: Austral summer definition, from what month to what month?

L104-L105:In the austral summer, you are removing air temperature data when the wind speed is less than 2 m/s. Is ERA reanalysis data not using the same time? For example, when the wind speed is weak, radiative cooling becomes stronger. The monthly mean air temperature may be somewhat biased value in the austral summer.

L148:Is it reasonable to apply the lapse rate of air temperature even near the snow surface?

And, sensor height changes due to snow accumulation. Are the heights of the sensor corrected in consideration of snow accumulation for long-term observational stations?

L155:Although 300 kg/m3 of fresh snow density is used, it varies greatly depending on the observational period. If the observation period is short, it will be low, and if it is long, it will be high due to densification. Do you calculate the snow accumulation with the daily height change? How much range do you think of fresh snow density?

L159: The height of AWS wind speed sensor were all 3m ? You extrapolate 3m of AWS to 10m of ERA. Isn't it better to interpolate ERA 10m to AWS 3m high?

Table2: It is described as monthly mean. What month is it?

Table3: How are Zonal wind and Meridional wind calculated from AWS wind data? I think you need a description.

L201-203:The meaning of this sentence is hard to understand.

L229: Is the ‘event’ is the number of days? So, AWS records it as a snowfall event about once every three days by referring to the observational period. In ERA, less times than AWS counts. As mentioned in the discussion, it seems that the snow deposition caused by snow drift is also counted.

L252: Is SKYBLU not SKYHI?

L258: SKIHI is not SKYBLU?

L275, FIG2: At THUR station, wind rose of AWS and ERA also looks different.

L318-319: Are the effects of evaporation, sublimation and snowmelt not considered?

L358-359:The meaning of this sentence is hard to understand.

4.2 Implications for ice core proxy calibration:I do not know the importance of this session. What criteria did you select? The snow depositional environment and the densification process after accumulation are also important. The ice core is recovered for various purposes. Is it the purpose of finding the best place for paleoclimate reconstruction? Without comparison with ice core data, it is not very persuasive.

Author Response

Our response to the reviewer's comments are in the attached file "author-coverletter-4357024.v1.docx"

Round  2

Reviewer 2 Report

It seems that you have responded to my comments as much as possible. But if I ask the same general comments, you will answer the same. So, I will finish the question for you.